# Recovery of Uranium by Se-Derivatives of Amidoximes and Composites Based on Them

**DOI:** 10.3390/ma14195511

**Published:** 2021-09-23

**Authors:** Eduard Tokar, Konstantin Maslov, Ivan Tananaev, Andrei Egorin

**Affiliations:** 1Institute of Chemistry, Far Eastern Branch of Russian Academy of Sciences, 690022 Vladivostok, Russia; users@local.ich.dvo.ru; 2School of Life Sciences, Far Eastern Federal University, 690090 Vladivostok, Russia; maslov.kv@dvfu.ru (K.M.); geokhi@mail.ru (I.T.)

**Keywords:** uranium, amidoximes, sorption, 1,2,5-oxadiazoles, selenium

## Abstract

An Se-derivative of amidoxime was synthesized for the first time as a result of the reaction of oxidative polycondensation of N’-hydroxy-1,2,5-oxadiazole-3-carboximidamide with SeO_2_: its elementary units are linked to each other due to the formation of strong diselenide bridges. The element composition of the material was established, and the structure of the elementary unit was suggested. The sorption-selective properties were evaluated, and it was found that the adsorbent can be used for the selective recovery of U (VI) from liquid media with a pH of 6–9. The effect of some anions and cations on the efficiency of recovery of U (VI) was estimated. Composite materials were fabricated, in which silica gel with a content of 35, 50, and 65 wt.% was used as a matrix to be applied in sorption columns. The maximum values of adsorption of U (VI) calculated using the Langmuir equation were 620–760 mg g^−1^ and 370 mg g^−1^ for the composite and non-composite adsorbents, respectively. The increase in the kinetic parameters of adsorption in comparison with those of the non-porous material was revealed, along with the increase in the specific surface area of the composite adsorbents. In particular, the maximum sorption capacity and the rate of absorption of uranium from the solution increased two-fold.

## 1. Introduction

The growing demand for uranium ore due to its widespread use in the nuclear industry is accompanied by the accumulation of solid and liquid radioactive waste containing uranium and its daughter decay products. The process of continuous release of uranium and its decay products to the biosphere occurs from rocks and parent rocks due to natural factors. This process is intensified during the mining and subsequent processing of uranium ore at nuclear industry facilities, thus creating risks of contamination of natural waters used in agriculture, as well as sources of drinking water [1]. Considering the highly toxic and radiotoxic hazards of uranium isotopes, characterized by a wide range of water-soluble forms in the biosphere, there is an urgent task to find methods for its selective recovery and concentration from natural and technological waters, including treating it as a valuable raw material.

A number of methods were developed over the past few decades for the recovery of uranium from aqueous solutions: extraction [2], membrane filtration [3], photocatalysis [4], chemical precipitation [5], electrocoagulation [6], adsorption [7], and others. However, most traditional methods have their disadvantages. In particular, despite the high efficiency of uranium extraction by membrane filtration, the main disadvantages of this method include the high energy costs required to restore the membranes and maintain high pressure [4]. The chemical deposition method is quite simple and economical, but it is often ineffective for reducing the concentration of uranium below the legally permissible levels and tends to form secondary wastes [5]. In addition, under conditions of working with high-salinity solutions, the production of a pure precipitate of uranium salts is complicated by the precipitation of salts of related metals, which requires an additional separation stage.

Among all the known methods, the most preferred is selective adsorption, which is free from most of the above disadvantages. Various natural and synthetic materials can be used as adsorbents. As examples of natural adsorbents, one can single out various types of aluminosilicates, such as kaolinite [8], montmorillonite [9], etc. The advantages of natural aluminosilicates include low cost, mechanical strength, and absence of negative impact on the environment so that these materials are often used to create migration barriers. However, low selectivity limits the practical use of aluminosilicates for the recovery of uranium isotopes.

The class of synthetic materials includes a large number of different adsorbents, which can be divided into inorganic and organic ones, and composites based on them. However, most of the known adsorbents have a number of limitations of application, such as a narrow range of the working pH of the solution [10], low kinetic parameters of radionuclide adsorption [11], and the use under conditions of low mineralization [11] due to the strong influence of competing ions.

Here, oxides and hydroxides of the following metals can be used as inorganic materials for the recovery of uranium from liquid media: Mn [12], Fe [13], Ti [14,15], Al [16], etc. They can be used for the recovery of uranium from various liquid media, including seawater in a wide pH range. The disadvantages of these sorbents include low adsorption capacity, as well as partial destruction, which leads to the desorption of uranium [17]. The authors of [10,18] suggested using titanosilicates, amorphous spherical granular phosphates, and titanium phosphatosilicates as adsorbents for the extraction of uranium isotopes. The disadvantage of the latter materials consists in their use for the recovery of just cationic forms of radionuclides, i.e., from aqueous media with pH less than 5.5.

Chitosan is a well-known adsorbent, which is used to recover uranium from mining wastewaters generated during the processing of uranium-containing ores. The sorption properties of chitosan towards uranium are determined by the presence of glucosamine and acetyl-glucosamine functional parts with hydroxyl and amino groups [19]. One of the representatives of this class of sorbent is the fibrous chitin–melanin glucan complex Mikoton obtained from higher fungi [20]. A number of studies [20,21] demonstrated the prospects of using various modifications of this sorbent for the selective recovery of a number of radionuclides such as Cs, Sr, Am, Pu, and U. A special feature of this sorbent consists in its fine-fiber structure characterized with high mechanical strength, thus providing an exceptionally developed surface, which allows it to be used under harsh sorption conditions. A large number of functional groups and the ionic permeability of the material contribute to a greater sorption capacity. However, despite the unique physical and chemical properties of the material, high sorption-selective characteristics of sorbents are attained at a pH of less than 5, which significantly limits the scope of their application.

Among organic adsorbents, materials with chelating functional groups, such as amino, sulfhydryl, carboxyl, amidoxime, and imidiacetate, are of the greatest interest due to their significantly high adsorption capacity for binding uranium, which is the result of ion exchange and complexation reactions [22,23,24].

Sorption materials that have an amphoteric ligand amidoxime with an acidic oxime and basic amino groups in their structure are of particular interest. This ligand allows the formation of stable chelates with a wide range of heavy metals, including Pb^2+^, Cu^2+^, Cd^2+^, Co^2+^, and Ni^2+^, as well as a five-membered chelate with U(VI), by providing lone-pair oxygen electrons at the oxime and amino groups to a positively charged metal [25].

In early studies of the use of amidoximes for metal binding in aqueous solutions, it was suggested that this class of materials was the most effective for recovering uranium from liquid media [26]. The authors of [27] showed the binding of uranyl ions in the uranyl–acetamidoxime complex with stability constants greater than 1 × 10^10^, which indicated high selectivity of the materials. In order to increase the sorption capacity, composite sorbents were synthesized by amidoximation of the following materials: polyethylene [28,29], carbon tubes [30], chitosan [31], and cellulose [32]. However, despite high selectivity towards uranium and high sorption capacity, their use in real conditions is limited by low hydro-mechanical strength [30] and solubility in alkaline media [31]. The use of polyurethane foam to increase the specific surface area leads to partial or complete blockage of the pores in the process of chemical modification [28,29]. Besides, the multi-step synthetic modification leads to the decrease in the degree of inoculation of amidoxime and spatial separation of chelate groups in the sorbent, which adversely affects the ability to bind uranyl ions and decreases the affinity. The materials comprising ordered three-dimensional coordination polymers consisting of metal cations (Al, Zn) and amidoxime radicals [33] are promising—they exhibit high sorption capacity.

To sum up, sorption materials based on amidoxime compounds are actively studied from the practical point of view, where they have proven to be highly effective for recovering uranyl ions from anthropogenic and natural liquid media with high contents of carbonates and sulfates, as well as seawater.

Despite significant advances in the production of polymer organic materials based on amidoximes and oximes, methods of production and properties of organic sorbents with an inorganic component of the Se type were not earlier published. In view of this, the objective of the present work was to synthesize new derivatives of 4-aminofufazane-3-carboxamidoxime, both in the pure form and as mechanically strong composites based on silica gel, through the introduction of heteroelements of high coordination capacity, which would facilitate the formation of polymer materials characterized with developed structure and high content of ion-exchange groups (amino and hydroxyl groups) and distinguished by an increased selectivity to uranium.

## 2. Materials and Methods

Selenium dioxide, acetic acid, ethanol, and methylene chloride of the especially pure grade were used to prepare the sorption material, as well as silica gel with a particle size of 35–70 mesh. In order to prepare working solutions simulators, metal salts of the chemically pure grade were used without additional purification—they were purchased from Nevareaktiv LLC (Saint-Petersburg, Russia). The radionuclide U-235 in the form of UO_2_(NO_3_)_2_ of the especially pure grade was used as a sorption element.

### 2.1. Preparation of Materials

The 4-aminofurazane-3-carboxamidoxime (compound **1**) was preliminarily synthesized by the following method [34]. An amount of 7.6 g of sodium nitrite (110 mmol) was gradually added to 140 mL of a solution of malononitrile (100 mmol) under intensive stirring at 20 °C, and then the mixture was cooled down to 5 °C. Thereafter, for 15–20 min, 3.8 mL of concentrated H_3_PO_4_ solution was added to it drop by drop under intensive stirring; the mixture was again stirred for 1.5 h and cooled down to 15–20 °C. Finally, 21 g of hydroxylamine hydrochloride (0.3 mol) was added to the mixture, and the stirring continued until the latter was completely dissolved. Thereafter, the mixture was alkalized to pH 10 using a solution of KOH (0.45 mol) and boiled for 2 h. The resulting solution was slowly cooled down to room temperature and acidified to pH 7 using a 30% H_3_PO_4_ solution. The resulting mixture was held at 0–5 °C for 12 h for crystallization, and the final compound was filtered and washing with cold water. The yield of compound **1** was 75–85%. Prior to use, compound **1** was recrystallized from an aqueous solution in the presence of activated carbon.

### 2.2. Synthesis of the Se-Derivative N’-hydroxy-1,2,5-oxadiazole-3-carboximidamide

A mixture of 1.43 g (10 mmol) 4-aminofurazane-3-carboxamidoxime 1 and Se(IV) oxide with a weight of 1.11 g (10 mmol) was boiled in acetic acid (30 mL) with a reverse refrigerator for 80–100 min under intensive stirring, a dark purple precipitate was formed as a result. At the end of the reaction, the mixture was cooled down to room temperature, and then the precipitate was separated from the mother liquor by a *blue ribbon* filter (JSC “ECOS-1”, Moscow, Russia). The precipitate was then washed sequentially with cold distilled water (three times), ethanol (once), and methylene chloride (once) to remove the unreacted components of the initial mixture. The resulting product was dried to a constant weight in a vacuum desiccator at 5–10 mm Hg above P_2_O_5_ at a temperature of 20–25 °C for 24 h. The final material, denominated as Se-init had a yield of 80–90% and comprised dark purple granules of an irregular shape with a grain size of 0.05–0.2 mm [35].

### 2.3. Synthesis of Composite Sorbents

In order to prepare composite samples, silica gel was added to a mixture of 1.43 g (10 mmol) of 4-aminofurazane-3-carboxamidoxime, 1.11 g (10 mmol) of selenium dioxide, and 30 mL of acetic acid during boiling. Boiling of the mixture lasted 80–100 min under intensive stirring. After completely falling out of the dark purple precipitate, the boiling was stopped, and the mixture was gradually cooled to room temperature and filtered. The resulting material was washed with cold water (three times), ethanol (twice), and methylene chloride to remove unreacted organic compounds. The final product was dried to a constant weight in a vacuum oven at 50 °C. The prepared materials comprised dark purple powders with a grain size of 0.05–0.2 mm. Three types of composite materials were fabricated; they were denominated as Se-35, Se-50, and Se-65, with a silica gel content of 35, 50, and 65 wt.%, respectively.

### 2.4. Study of Sorption Characteristics under Static Conditions

In the sorption experiments, initially, the U(VI) salt (nitrate) was used. However, to simplify the description, hereinafter, U(VI) and its possible forms of existence will be marked as just uranium.

The sorption properties of the fabricated materials were investigated using solutions of pH of 2–10 containing uranium with the ratio V/m = 1000 mL g^−1^ (*V* is the volume of the model solution, *m* is the weight of the sorbent). The phase contact time was 48 h under continuous stirring on an orbital shaker with an amplitude of 10 mm and a rotation speed of 150 rpm. Prior to the start of the uranium recovery experiment, the sorbent samples were held for 24 h in the model solutions that did not contain uranium.

At the end of the experiment, the model solution was separated from the sorbent by a *blue ribbon* filter, and the residual uranium^+^ content was analyzed by the spectrophotometric method using Arsenazo III at a wavelength of 656 nm [36]. The results obtained were used to further calculate the extraction efficiency (S%) and the uranium distribution coefficients (*K_d_*, mL g^−1^):

The extraction efficiency was calculated according to Equation (1):(1)$$S=\left(1-\left(\frac{C_{e}}{C_{i}}\right)\right)\times 100$$

The value of the uranium distribution coefficient (mL g^−1^) was calculated using Equation (2):(2)$$K_{d}=\frac{C_{i}-C_{e}}{C_{i}}\times \frac{V}{m}$$
where *C_i_* is the initial concentration of uranium in the solution (mg L^−1^), *C_e_* is the equilibrium residual concentration of uranium in the solution (mg L^−1^), *V* is the volume of the solution (mL), and *m* is the weight of the sorbent sample (g). Additionally, at the end of the experiment, the equilibrium pH of the solution was measured, followed by the determination of the zero charge point (pHpzc) by a graphical method.

The effect of the presence of cations and anions in the solution on the static exchange capacity was evaluated in the presence of solutions of metal chlorides (K^+^, Na^+^, Mg^2+^, Ca^2+^, Co^2+^, and Ni^2+^) or sodium salts (HCO_3_^−^, NO_3_^−^, Cl^−^, SO_4_^2−^, and PO_4_^3−^), respectively. Solutions with the following concentration of the corresponding cations or anions were used: 0.001; 0.001; 0.01; 0.1 mol L^−1^. The static exchange capacity (SEC, mg/g) was calculated according to the following formula:(3)$$SEC=\frac{\left(C_{i}-C_{e}\right)\times V}{m}$$
where *V* is the volume of the solution (L).

The nature of the adsorption process was estimated from the uranium adsorption isotherms, which were obtained by contacting a series of samples of a given weight with the model solutions of pH 6 and 8 containing a certain amount of uranium ^+^. The standard Langmuir Equation (4) was used to describe the sorption isotherm:(4)$$\Gamma =G_{max}\times \frac{K_{l}\times C_{e}}{1+K_{l}\times C_{e}}$$
where *G_max_* is the maximum sorption value (mg g^−1^), *K_l_* are the constants of the adsorption equilibrium characterizing the energy of the adsorbent–adsorbate bond (L g^−1^).

The experimental data were approximated by means of the specified equations using the SciDAVis software (version 1.23, scidavis.sourceforge.net, accessed on 15 September 2021).

The kinetic characteristics of the sorption process were evaluated using a model solution with pH 6 containing 0.1 mol L^−1^ of NaNO_3_ and 30 mol L^−1^ of uranium. The adsorbents were brought into contact with the model solution, V/m = 200 mL g^−1^, and, after a certain period, aliquots of the solution were taken, and the residual content of uranyl ions in the solution was determined.

The obtained kinetic curves in S_(t)_—t coordinates were processed by the pseudo-first-order (Equation (5)) and pseudo-second-order (Equation (6)) kinetic equations.
(5)$$A_{t}=A_{e}\times \left(1-exp^{-k_{1}\times t}\right)$$
(6)$$A_{t}=\frac{k_{2}\times A_{e}{}^{2}\times t}{1+k_{2}\times A_{e}\times t}$$
where *A_t_* is the amount of adsorbed uranium at time *t* (mmol g^−1^), *A_e_* is the equilibrium uranium adsorption (mmol g^−1^), *t*—time (min), and *k_1_* and *k_2_* are the pseudo-first-order and pseudo-second-order constants, respectively.

The kinetic parameters of the adsorption process in solutions with pH 6 and 8 were comparable; therefore, this work presents the results obtained using solutions with pH 6.

The assessment of the hydro-mechanical strength (HMS) of the prepared sorbents was carried out as follows: a suspension of the sorbent with a weight of 1 g and grain size of 0.1–0.2 mm was mixed with 50 mL of distilled water for 24 h. Thereafter, the sorbent was separated from the solution, dried to a constant weight at 105 °C, sieved again, and weighed on an analytical balance with an accuracy of 0.001 g. The degree of destruction of the sorbent sample with a grain size of 0.1–0.2 mm was estimated by the ratio of the weight remaining after the mixing to the initial weight of the sample.
(7)$$HMS=\frac{m_{0}-\;m_{1}}{m_{1}}\times 100\%$$
where *m*_0_ is the initial weight of the sorbent with a grain size of 0.1–0.2 mm, *m*_1_ is the weight of the sorbent that retained the grain size of 0.1–0.2 mm after mixing and sieving.

### 2.5. Equipment

The infrared spectra (IRS) of the samples were recorded using a Spectrum 1000 spectrometer (Perkin Elmer, Waltham, MA, USA) in KBr tablets. X-ray diffraction patterns were recorded using an Advance D8 (Bruker, Billerica, MA, USA) device, with CuK_α_-radiation in the angle range of 2° < 2θ < 90° in the point-by-point scanning mode. The maximum deviation of the position of the reflexes, as determined through NIST SRM 1976, was less than 0.01° 2θ.

Nuclear magnetic resonance (^13^C NMR) spectra of the polymers in the solid phase were recorded using a Bruker Advance AV-300 device (Billerica, MA, USA) with a proton resonance frequency of 300 MHz. The method of magic-angle spinning (MAS) was used to record the spectra. Tetramethylsilane was used as a standard for carbon nuclei, whereas the chemical shift (CS) zero was set in a separate experiment. The error in determining the CS did not exceed 1–2 ppm, depending on the resolution of the peak. The spectra were recorded at a temperature of 300 K.

The X-ray photoelectron spectroscopy (XPS) spectra were obtained using an ultrahigh vacuum photoelectron spectrometer manufactured by Specs (Berlin, Germany) with a hemispherical electrostatic energy analyzer PHOIBOS (Specs, Berlin, Germany). An X-ray tube with a magnesium anode (MgKa-1253.6 eV, Specs, Berlin, Germany) was used as the radiation source. The pressure in the chamber during the measurement was 5 × 10^−7^ mbar. The energy scale was calibrated according to the carbon level C 1s, the energy of which was assumed to be 285.0 eV. The transmission energy of the analyzer was 50 eV; the step at scanning the survey spectra was 1.0 eV, while for the detailed spectra, it was equal to 0.1 eV. At the sample preparation stage, the investigated materials were pressed onto a 1 cm × 1 cm sticky conductive tape.

The high-resolution scanning electron microscopy study was performed using a HITACHI TM 3000 device (Hitachi, Tokyo, Japan) at accelerating voltages of 5–15 kV and a beam current of I ≈ 100 pA. The device was equipped with an accessory for energy dispersion analysis (EDA) by Bruker (Berlin, Germany).

The specific surface area and the pore size of the material were determined by the method of low-temperature nitrogen adsorption using an Autosorb IQ device by Quantachrome Instruments (Boynton Beach, FL, USA). The calculation was performed according to the BET method (the theory of Brunauer–Emmett–Teller).

The element content in the model solution was evaluated by the atomic absorption flame spectroscopy using a Thermo Solar AA M6 device (Thermo Electron Corporation, Waltham, MA, USA). The surface morphology was studied by scanning electron microscopy (SEM) using a Carl Zeiss CrossBeam 1540-XB instrument (ZEISS Microscopy, Munich, Germany).

The content of uranyl ions in the solution was determined by the spectrophotometry method using a Shimadzu UV-1800 spectrophotometer (Kyoto, Japan).

## 3. Results and Discussion

### 3.1. Physiochemical Properties

According to the objective of the present work, the materials based on amidoxime functional groups, in particular, the Se-derivative of N’-hydroxy-1,2,5-oxadiazole-3-carboximidamide (Se-init), were studied as highly effective sorbents for uranium recovery. The choice of Se was determined by the mild selective oxidizing properties of SeO_2_ towards N’-hydroxy-1,2,5-oxadiazole-3-carboximidamide. The method of preparation is based on the reaction of oxidative polycondensation of N’-hydroxy-1,2,5-oxadiazole-3-carboximidamide with SeO_2_ in the presence of polar organic solvents followed by the formation of a polymer structure, whose elementary units are linked to each other due to the formation of strong diselenide bridges. This original method offers a very simple implementation by a single-stage reaction, as well as the use of an organic radical with a large number of reactive groups (hydroxyl and oxime) that remain in the final product. A number of physical–chemical methods were used to determine the molecular structure of the resulting compound. In particular, Figure 1 shows the IR spectrum of Se-init.

The bands observed in the spectrum indicate that the initial compounds are built into the general structure and preserve functional radicals. An intense absorption band is observed at 3454 cm^−1^, which corresponds to the O–H vibrations at the oxime group. The absorption band at 3311 cm^−1^ corresponds to N–H vibrations. The absorption band at 1623 cm^−1^ corresponds to C=N vibrations and the one at 1357 cm^−1^ to vibrations of C–N bonds. In the Se-init spectrum, one observes narrowing of absorption bands at 3311 and 3454 cm^−1^, which indicates the absence of expressed hydrogen bonds. The specific feature of the Se-init spectrum (Figure 1b) consists in the presence of peaks at 412 and 489 cm^−1^ corresponding to vibrations of Se–Se and N–Se bonds, respectively. In addition, an important feature consists in the presence of the absorption bands at 1548 cm^−1^, corresponding to the Se=C-N bond vibrations, and 1105 cm^−1^, corresponding to the C–NO bond vibrations.

The X-ray diffraction analysis showed the presence of peaks corresponding to the initial N’-hydroxy-1,2,5-oxadiazole-3-carboximidamide, which indicated the complete binding of the functional organic part with partial preservation of the original structure (27 Å—Figure 2, X-ray Pattern b). The syngony and the spatial group P 3(1)21 were established—a structure similar to gray selenium (β-Se) but different from it (Figure 2, X-ray pattern b). In addition, the Se-init X-ray pattern shows signals corresponding to the Se phases that were not earlier found in the published data.

The presence of the crystal structure can be indirectly corroborated by the images of scanning electron microscopy (Figure 3). One can see that the morphology of the Se-init surface is represented by clearly expressed crystallites of irregular structures and sizes.

The material was analyzed by the method of solid-state NMR on ^13^C and ^1^H nuclei. In the ^13^C spectrum (Figure 4a), there is a clearly expressed signal at 138 ppm corresponding to carbon at the oxime group—C=NOH, as well as signals of 146 ppm corresponding to the -HN–C= and 154 ppm typical for =C–C= (Table 1). In the PMR spectrum (Figure 4b), there are signals of 1.5 ppm corresponding to the proton at aliphatic C–NH, and 7.1 ppm corresponding to H–N(C)–Se.

The additional PMR analysis in the liquid phase was performed: it showed the presence of an additional signal with an intensity of 3 at 2.73 ppm, which probably corresponded to three acetylene protons. Figure 5 shows the XPS spectra of the prepared material. In order to obtain a more accurate understanding of the structure of the compound, the material was pre-washed with distilled water and ethanol alcohol to remove the synthesis by-products.

The XPS spectrum shows the presence of a C 1s peak (Figure 5b) around 285.0 eV, which corresponds to the aliphatically bound carbon at the position CH_2_–CH-; the peak of 286.9 eV corresponds to the tri-substituted carbon NH_2_–CH=N–OH. A signal of energy of 288.5 eV corresponds to the groups C–N/C=N. In the region corresponding to N 1s (Figure 5c), there are peaks with binding energies of 399.2 eV typical for triazine-type nitrogen C–N–OH and of 400.4 eV corresponding to thiadiazole-type nitrogen, probably bound to the selenium atom (Se–NH–C). The signals of the O 1s peak (Figure 5d) with binding energies of 533.6 eV and 531.2 eV correspond to an oxygen atom with surroundings of NH_2_–C=N–O–N and N–O–H, respectively. The position of the Se 3p peak (Figure 5e) of the binding energy of 164.0 eV is typical for tetravalent selenium, probably bound to the second selenium atom by a double bond (>Se=Se<). The spectrum also contains a signal at 169.3 eV corresponding to a selenium atom with the C–NH–Se surrounding.

Based on the performed physical and chemical analysis, we suggested the mechanism of binding of compound **1** (Figure 6) to selenium dioxide and suggested the molecular structure of the resulting material (Figure 6).

It was found that, as a result of the polycondensation reaction, the degree of oxidation of Se (IV) is preserved with the formation of a strong double bond >Se=Se< bound by functional amidoxime radicals.

During the synthesis, compounds 2 and 3 remained in the dissolved phase and were then completely removed by repeated washing with distilled water and organic solvents, as evidenced by the results of the analysis of the material (Table 1).

Table 1 shows the data calculated for an elementary unit of Se-init (Figure 6). The experimental percent values of the elements match the theoretically calculated ones with insignificant deviations, which is probably related to the presence of non-removable monomeric or oligomeric structures.

### 3.2. Sorption Selective Properties

Figure 7 shows the experimental results of the determination of the zero charge point (pH_pzc_) using a NaNO_3_ solution of a concentration of 0.01 mol L^−1^. The results are shown in the form of a dependence of the equilibrium pH of solutions after contact with a sorbent on the initial pH of these solutions. The bend point on curve 3 corresponds to the zero charge point that is the value of the solution pH, at which the sorbent surface is uncharged. In our case, the pH_pzc_ value is equal to 4.3. Below this value of solution pH, the sorbent surface is charged positively, which prevents uranium adsorption due to electrostatic repulsion. Above pH_pzc_, the sorbent surface is charged negatively, which positively affects the efficiency of recovery of uranium and its hydrated cationic forms.

The results of the determination of pH_pzc_ are in good agreement with the dependence of the efficiency of uranium recovery on the solution pH (Figure 8). The solution pH correction was carried out using solutions of HNO_3_ and NaOH. In acidic solutions, low recovery efficiency is related to positively charged sorbent surface as, probably, to the predominant reaction of protonation.

The form of uranium presence in a solution depends significantly on the solution pH. In solutions with pH above 4, one observes the increase in the uranium recovery due to strong binding by the adsorbent negatively charged surface and predominant cationic hydrate forms of uranium. The dependence of the composition of the hydrated uranium cation with the pH increase (from 4 to 8) is expressed in the row UO_2_OH^+^ → (UO_2_)_2_(OH)_2_^2+^ → (UO_2_)_3_(OH)^5+^ [37]. The recovery efficiency maximum is attained at pH 8, with a subsequent decrease at the transition to the alkaline range. The decrease in the efficiency of uranium recovery in solutions with pH above 8 is probably related to the increased formation of the anionic form of uranium UO_2_(OH)^3−^ [37] and the uranium adsorption on the flask walls, which was found in the control experiment without the sorbent (curve 3, Figure 8). The high efficiency of the uranium recovery is explained by the presence of a large number of sorption sites forming at polymer formation as a result of polycondensation of cyclic molecules through the ˃Se=Se< bond.

Table 2 shows the values of the sorption-selective properties of the sorbent, depending on the initial pH of the solution. According to the obtained data, these materials can be recommended for the recovery of uranium from neutral and slightly alkaline solutions. A change in the value of pH of the model solution as compared to the initial value was observed. This was probably the result of deprotonation of the =N–O–H group during the adsorption.

Figure 9 shows a diagram of the dependence of the SEC uranium values on the concentration of cations in the solution. It was found that double-charged cations reduce the SEC value to a greater extent than single-charged cations, which is explained by the electrostatic interaction, as well as due to the larger number of occupied adsorption sites, which is associated with an increase in the equivalence factor of multiply charged elements. The sequence of cations by the degree of increasing negative impact on the value of SEC is as follows: Na^+^ ≤ K^+^ < Mg^2+^ ≤ Ni^2+^ ≤ Ca^2+^ ≤ Co^2+^. However, it is worth mentioning that despite the decrease in SEC values at increasing the concentration of competing cations, the efficiency of the uranium recovery exceeds 75%, regardless of the competing ions.

The uranium extraction efficiency in the presence of other (competing) cations at pH 6 and 8 is generally comparable to each other. For this reason, this experiment was carried out only at pH6. The most negative effect on the uranium extraction efficiency is exerted by anions, the effect of which strongly depends on the solution pH.

Uranium undergoes hydrolysis and has the affinity to complexation: this effect is clearly expressed in neutral and slightly alkaline media, which significantly complicates its recovery by sorption. The presence of various anions in the solutions, such as bicarbonates or sulfates, can significantly reduce the efficiency of uranium extraction. Experiments were carried out to estimate the negative effect of anions.

Figure 10 shows the curves of the dependence of the SEC uranium values on the type and concentration of some anions that may be present in natural and technological liquid media.

It was found that the anions negatively affected the SEC value to varying degrees. The sequence of anions according to the degree of increasing negative impact on the value of SEC is as follows: NO_3_^−^ < PO_4_^3−^ < Cl^−^ < HCO_3_^−^ < SO_4_^2−^. The greatest negative effect is provided by bicarbonate and sulfate ions, which is explained by the formation of stable anionic complexes of the type [UO_2_(SO_4_)_3_]^4^^−^, [(UO_2_)_2_CO_3_(OH)_3_]^−^, [UO_2_(CO_3_)_3_]^4^^−^, [UO_2_(CO_3_)_2_]^2^^−^ and [UO_2_(CO_3_)_2_(H_2_O)_2_]^2^^−^ [38], the probability of their formation increases with the pH growing. However, when the anion concentration is equal to 0.01 mol L^−1^, one observes less than a two-fold decrease in the adsorption of uranium, which is also associated with the formation of complex ions such as [UO_2_NO_3_]^+^, [UO_2_PO_4_]^–^, [UO_2_Cl]^+^ UO_2_Cl_2_^0^, etc. [39]. The weakening of the sorption properties in alkaline solutions is related to the partial oxidation of the organic radical, accompanied by the destruction of the sorbent.

The Se-init after the sorption of uranyl ions was analyzed by the XPS spectroscopy: the results showed the presence of U 4f signals with binding energies of 392.8 and 382.0 eV, which indicated the binding of uranyl ions in the sorbent due to complexation and partial ion exchange. Figure 11 illustrates the structure of the binding of uranyl ions by Se-init. After the deprotonation reaction of the –OH group, the binding of the radionuclide can proceed according to the M1 mechanism in acidic or neutral media, as well as by the M2 mechanism (Figure 11) in slightly alkaline and alkaline media due to binding of the oxygen and nitrogen atoms at the oxime group.

The preservation of mechanical stability is an important requirement for sorption materials created for long-term use in sorption columns. This requirement can be fulfilled by developing composite materials by depositing them on a mechanically stable matrix.

In the present work, silica gel was used as such a matrix: it was modified with a polymer-based on the Se-derivative of amidoxime. Table 3 shows the surface characteristics of the resulting materials: the data were obtained using nitrogen adsorption and subsequent calculation according to the BET equation. It was shown that, unlike the initial material (Se-init), the composite materials have a more developed surface, which increases proportionally to the increase in the weight content of silica gel.

Figure 12 shows the adsorption isotherms obtained in the solutions with pH 6 (Figure 12a) and pH 8 (Figure 12b), which can be attributed to the L-type [40], thus indicating a high affinity of the adsorbents to uranium. Besides, the highest SEC is attained at pH 8, which is probably related to peculiarities of the ionic form of uranium in solution: in particular, at pH 8, the predominant forms are UO_2_(OH)_3_^–^ and (UO2)_3_(OH)_5_^+^, while at pH 6—UO_2_(OH)^+^, (UO_2_)_2_(OH)_2_^2+^, UO_2_(OH)_2_^0^, and (UO2)_3_(OH)_5_^+^, respectively [37].

Table 4 shows the values of *G_max_* and the adsorption equilibrium constant *K_l_* obtained by approximating the experimental points. The coefficient of determination exceeds 0.95, which indicates that the chosen model is correct. For the composite sorbents, the values of the adsorption equilibrium constant are quite close, which indicates the preservation of the adsorption mechanism. The adsorption equilibrium constants obtained for the composites have lower values in comparison with Se-init due to the reduced content of the sorption-active component (Se-derivative of 4-aminofurazane-3-carboxamidoxime). The highest values of the maximum adsorption were obtained for Se-50 and Se-65, which could be related to the greater availability of adsorption sites; however, we do not exclude the adsorption of uranyl ions on silica gel.

The highest values of the maximum adsorption were obtained for Se-50 and Se-65, which could be related to the greater availability of adsorption sites and also with partial adsorption of uranium on silica gel. In order to evaluate the silica gel effect on the composite’s sorption characteristics, a control experiment was carried out. In the control experiment, the efficiency of uranium adsorption with unmodified silica gel was evaluated.

Figure 13 shows the diagram of uranium sorption from model solutions with pH 6 and 8 on synthesized materials and pure silica gel used in the fabrication of composite materials. It is known that various types of silica gels are capable of adsorbing heavy metal ions from liquid media [41,42]. It should be noted that unmodified silica gel cannot efficiently extract uranium since ion-exchange hydroxyl groups in silica gel do not exhibit high selectivity to uranium in the presence of other metal ions and anions. In addition, the binding of uranium by embedding it in the crystal lattice of silica gel is significantly limited due to the low sorption capacity of the synthetic mineral. The efficiency of uranium extraction from the model solution does not exceed 50%; therefore, unmodified silica gel is not an effective sorbent. However, its use as a matrix for composite materials (along with Se-init) promotes the improvement of sorption properties, which is reflected in the increase in the efficiency of recovery (Figure 13) and SEC (Table 4), relatively to Se-init.

We evaluated the kinetic characteristics of the adsorption process on the composite sorbents. Figure 14 shows the kinetic curves of the sorption of uranium from the model solution in semi-logarithmic coordinates (Figure 14a). The kinetic curves indicate that the adsorption process on the composite sorbents proceeds at a higher rate, which is also associated with greater availability of the adsorption sites as compared with the non-composite sorbent. An increase in the specific surface area of composite materials leads to a decrease in the time to reach equilibrium adsorption, which is 10–20 min, while the adsorption efficiency exceeds 95%. A significant increase in the time to reach adsorption equilibrium on the Se-init sample is probably associated with the specific features of the adsorption process in the presence of uranium in various ionic forms. In addition, the diffusion process is important in the uranium-binding rate on a dense surface of a non-porous material, which in this case is likely to be limiting.

Figure 15 shows high-resolution SEM images for Se-init and the composite sorbents. Unlike Se-init, whose structure is represented by pronounced crystallites, the surface of the composite sorbents is relatively homogeneous and contains small particles of presumably the sorption-active component. The surface distribution of the sorption-active component indirectly corroborates the fact that the adsorption sites of the composite sorbent can have greater accessibility in comparison with the Se-init material, some of the sorption sites of which are located in the bulk of dense crystallites and therefore do not participate in the exchange.

Table 5 shows the parameters calculated for the pseudo-first and pseudo-second order equations (Figure 14b). According to the determination coefficients, the kinetic adsorption curve of uranium for Se-init is better described by the pseudo-first-order equation, as opposed to the composite adsorbents. The increase in the reaction order for the composite sorbents could also indicate the increase in the availability of the adsorption sites and the predominance of the chemisorption process of uranium binding in comparison with the non-composite sorbent. It is also worth mentioning that the pseudo-second-order equation constant *k_2_* differs for different composite sorbents. For the sorbent Se-65, the calculated *k_2_* has the lowest value, which is associated with a reduced content of the sorption-active component affecting the kinetic characteristics.

Table 6 shows the results of testing the hydro-mechanical strength of the fabricated materials. One can note that at a transfer to Se-35, the material loses stability rather sharply, which is probably related to an excess of the sorption-active component washed off from the surface of the matrix during the stirring. The latter is corroborated by the fact that Se-init is mostly peptized in the model solutions, which complicates working with it.

## 4. Conclusions

To sum up, a new approach to the synthesis of sorption materials containing amidoxime functional groups in their structure was developed. In particular, the Se-derivative 4-aminofurazane-3-carboxamidoxime was synthesized by polycondensation of individual substances in the presence of polar organic solvents. The molecular structure of the elementary unit of the initial Se-derived amidoxime was established, which indicates the formation of strong bridging bonds: >Se=Se<.

The sorption properties of the material under static conditions in the presence of both various cations and anions were characterized. The materials manifest mechanical and chemical stability, as well as high sorption-selective characteristics towards uranium in the pH range of 6–9.

In order to increase the kinetic parameters of ion exchange, the composite materials were fabricated based on the initial Se-derivative of amidoxime using silica gel as a matrix. The values of the maximum static exchange capacity in neutral and slightly alkaline model solutions were established: they equaled to 370 mg g^−1^ (pH-6) and 270 mg g^−1^ (pH-8) for Se-init and to 620 mg g^−1^ (pH-6) and 760 mg g^−1^ (pH-8) for Se-50.

It was found that, along with the increase in the specific surface area of sorbents, the kinetic parameters of the uranium sorption increase by order of magnitude as compared to the non-porous material.

Thus, the fabricated materials can be recommended for application in the recovery and concentration of uranium from technological and natural slightly alkaline waters with high mineralization.

## Figures and Tables

**Figure 1 materials-14-05511-f001:**
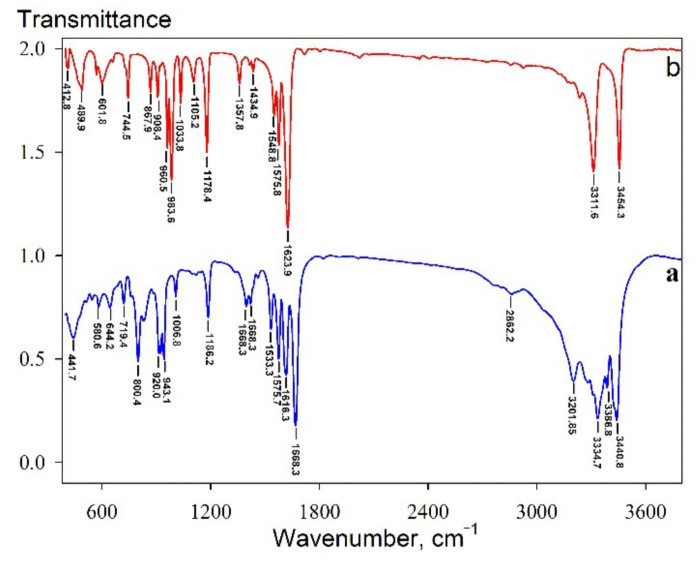
IR spectrum: (**a**) 4-aminofurazane-3-carboxamidoxime; (**b**) Se-init.

**Figure 2 materials-14-05511-f002:**
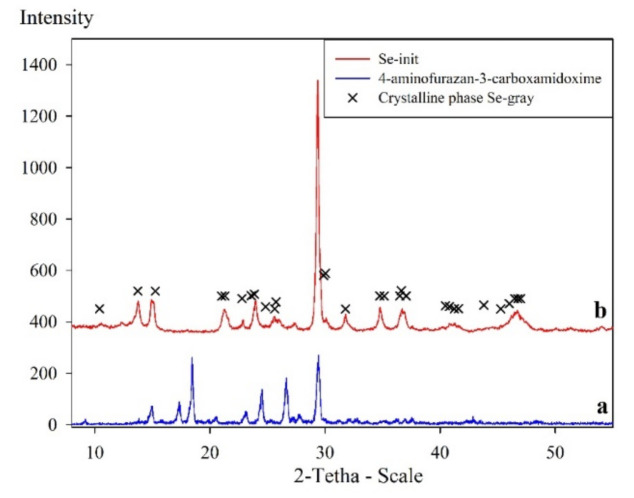
X-ray images of: (**a**) 4-aminofurazane-3-carboxamidoxime; (**b**) Se-init.

**Figure 3 materials-14-05511-f003:**
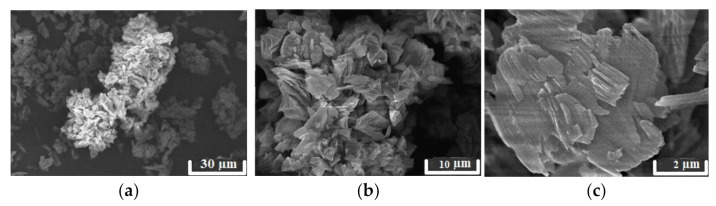
Se-init SEM images at different resolutions: (**a**) ×1500; (**b**) ×2500; (**c**) ×5000.

**Figure 4 materials-14-05511-f004:**
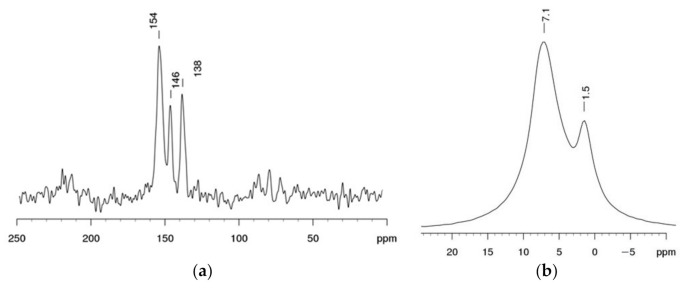
Se-init spectra: (**a**) ^13^C NMR, (**b**) PMR.

**Figure 5 materials-14-05511-f005:**
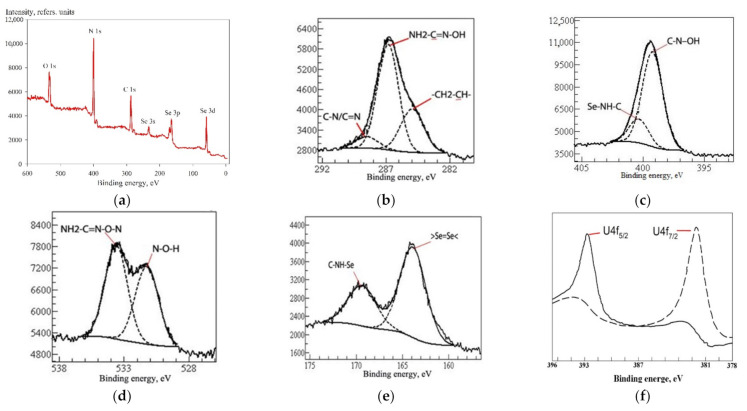
Se-init XPS spectra: (**a**) general view; (**b**) C 1s, (**c**) N 1s, (**d**) O 1s, (**e**) Se 3p, (**f**) U 4f.

**Figure 6 materials-14-05511-f006:**
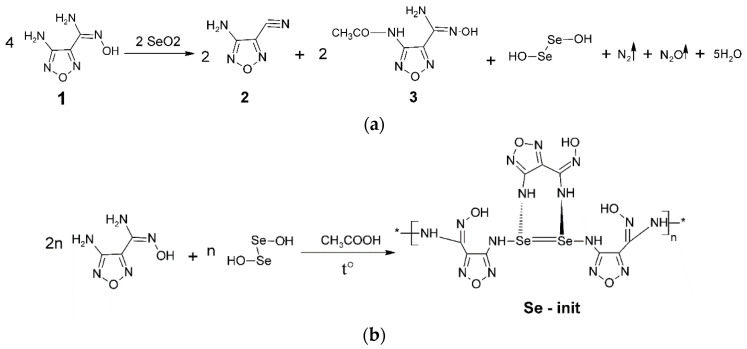
Mechanism for producing Se-init; (**a**) first stage; (**b**) second stage.

**Figure 7 materials-14-05511-f007:**
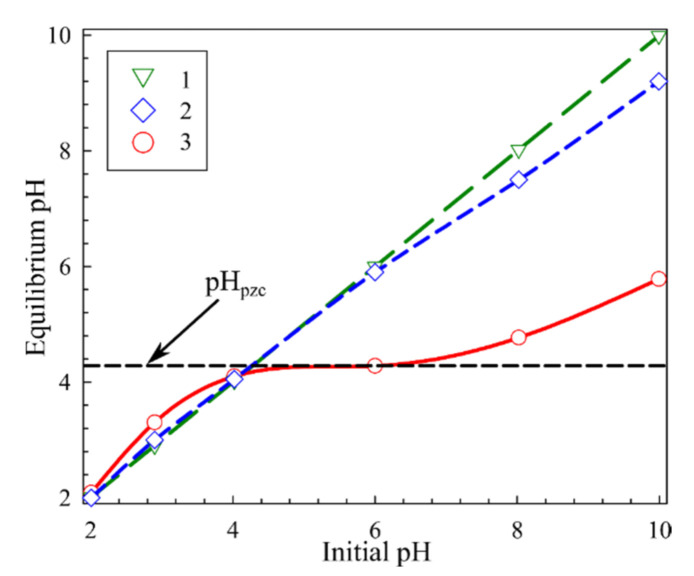
Point of zero charge (pH_pzc_) of the sorbent, 1—initial pH solution, 2—blank experiment (pH changing without sorbent), 3—changing the pH solution after contact with sorbent, within 24 h.

**Figure 8 materials-14-05511-f008:**
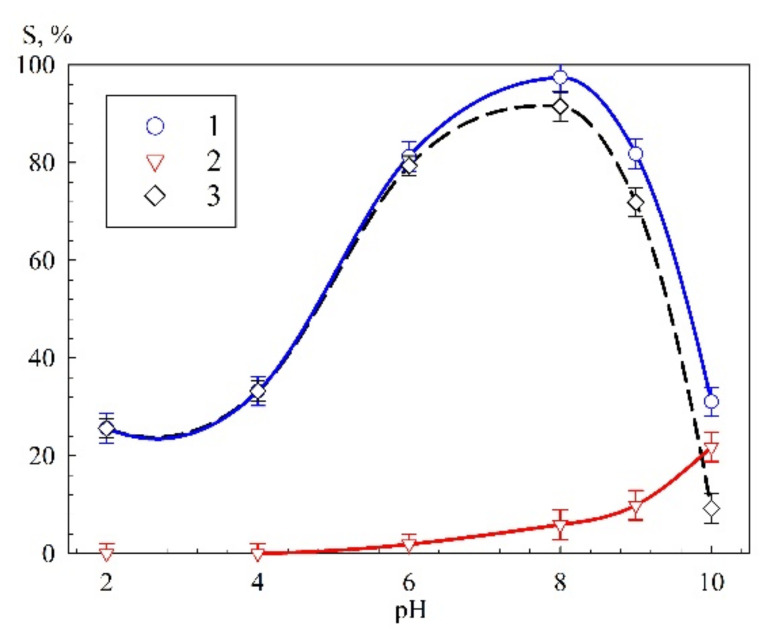
Dependence of the uranium sorption on pH of the model solution, 1—Se-init, 2—blank experiment, 3—adsorption efficiency taking into account adsorption on the flask walls.

**Figure 9 materials-14-05511-f009:**
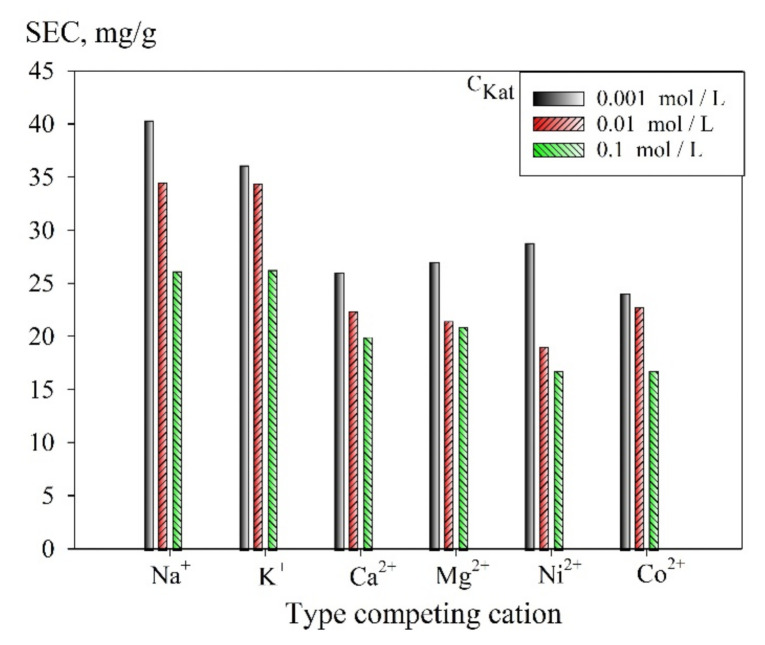
Diagram of the dependence of SEC (uranium) values on the type and amount of competing cations (V/m = 1000 mL g^−1^, pH 6), obtained at 25 °C.

**Figure 10 materials-14-05511-f010:**
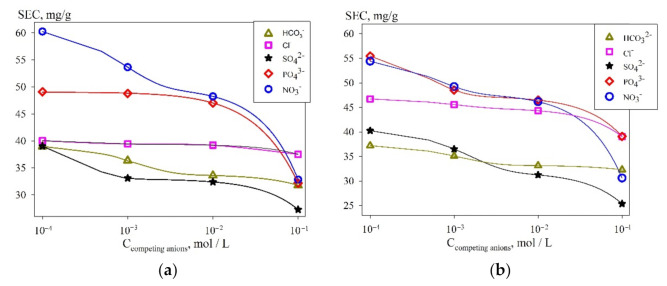
Dependence of SEC (uranium) values on the type and concentration of competing anions in a solution with (**a**) pH 6; (**b**) pH 8, obtained at 25 °C.

**Figure 11 materials-14-05511-f011:**
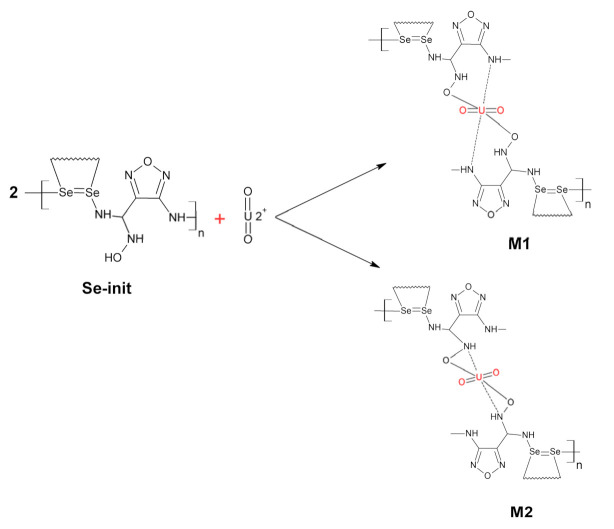
Scheme of possible structure of binding of uranium to Se-init.

**Figure 12 materials-14-05511-f012:**
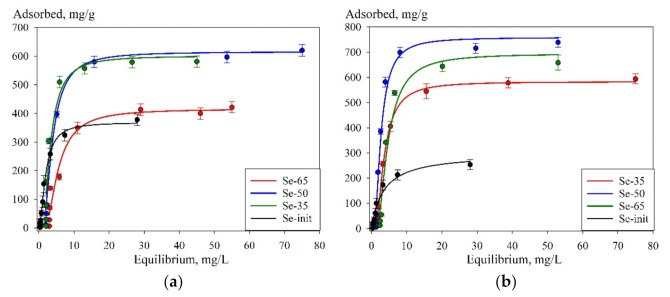
Sorption isotherms of uranium ((**a**)—pH 6, (**b**)—pH 8) and approximation of the experimental values by the Langmuir equation, obtained at 25 °C.

**Figure 13 materials-14-05511-f013:**
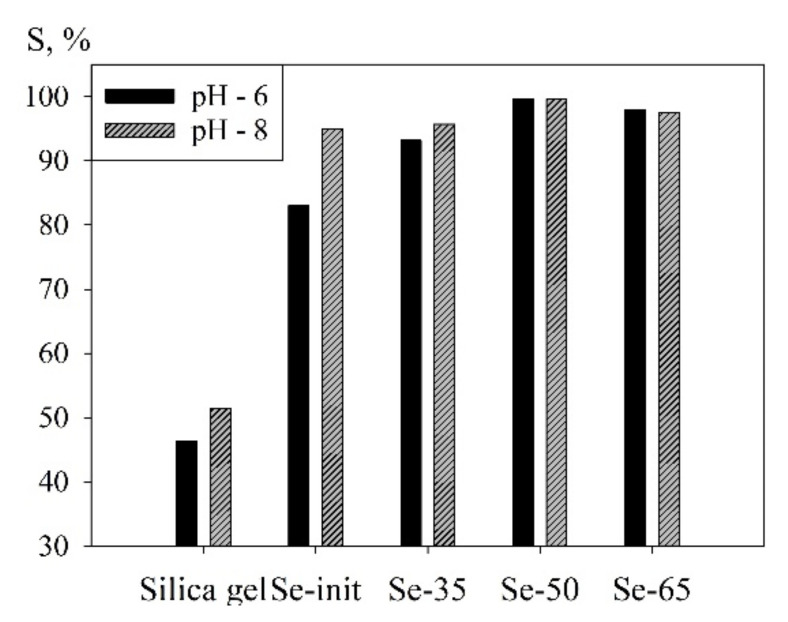
Sorption of uranium from model solutions, a—pH 6, b—pH 8, using various materials.

**Figure 14 materials-14-05511-f014:**
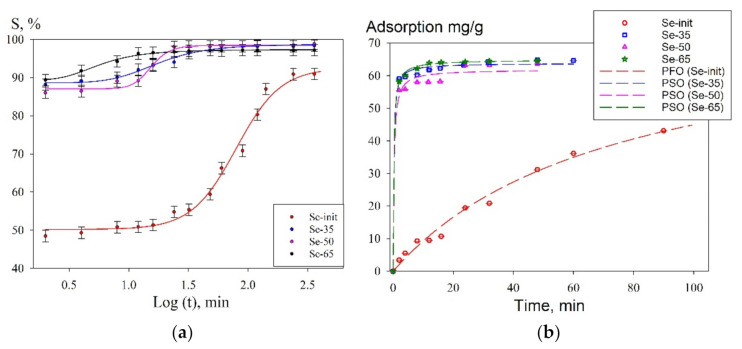
The recovery of uranium from the model solution under static conditions at 25 °C (V/m = 200 mL g^−1^); (**a**)—kinetic curves in semi-logarithmic coordinates; (**b**)—curves of the adsorption dependence (mmol g^−1^) on time.

**Figure 15 materials-14-05511-f015:**
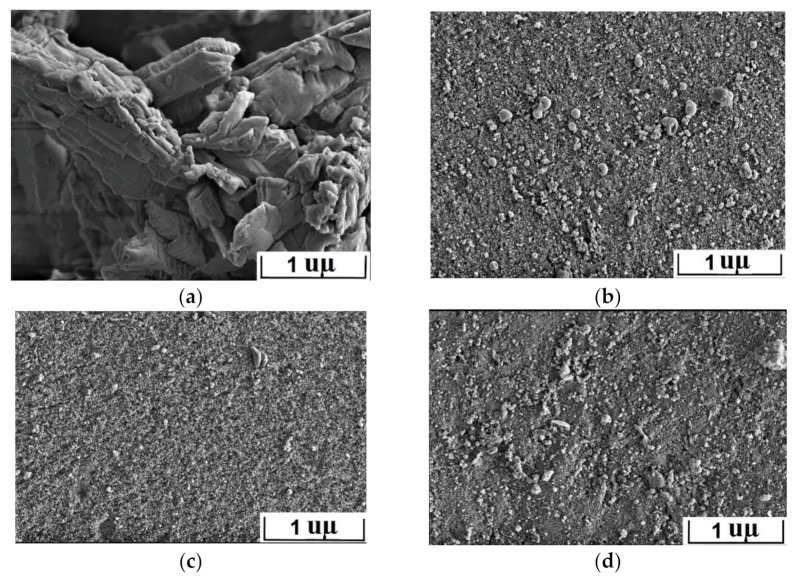
SEM images of the surface: (**a**) Se-init; (**b**) Se-35; (**c**) Se-50; (**d**) Se-65.

**Table 1 materials-14-05511-t001:** Se-init Surface Element Analysis.

Analysis Method	C, at.%	N, at.%	O, at.%	Se, at.%
Theoretical	29.97	46.70	16.67	6.66
EDS Analysis	29.41	44.38	17.94	8.28
XPS	33.80	45.20	16.10	6.85
REM	31.88	43.97	17.05	6.50

**Table 2 materials-14-05511-t002:** Dependence of the uranium^+^ adsorption parameters on the pH of the medium.

pH Initial	*K_d_* × 10^−3^, cm^3^ g^−1^	SEC, mg g^−1^
2	0.3	3.0
4	0.5	4.3
6	5.0	9.7
8	37.9	15.8
9	4.2	9.1
10	0.5	2.9

**Table 3 materials-14-05511-t003:** Results of the study of the surface of materials by the method of nitrogen absorption.

Parameter	Si-Init	Se-35	Se-50	Se-65
Silica gel content, wt.%	0	35	50	65
Pore volume, cm^3^ g^−1^	0.05	0.45	0.51	0.65
Specific surface area, ml g^−1^	2	210	243	298
Specific pore size, nm	1.21	12.1	12.1	12.1

**Table 4 materials-14-05511-t004:** Constants of the Langmuir equation calculated after approximating the experimental data.

	Parameter	Se-Init	Se-35	Se-50	Se-65
pH-6	G_max_	370 ± 20	420 ± 20	620 ± 20	600 ± 30
	K_l_	0.24 ± 0.05	0.14 ± 0.07	0.14 ± 0.05	0.14 ± 0.07
	R^2^	0.98	0.96	0.98	0.94
pH-8	G_max_	270 ± 10	580 ± 20	760 ± 30	690 ± 20
	K_l_	0.30 ± 0.07	0.19 ± 0.05	0.18 ± 0.07	0.21 ± 0.09
	R^2^	0.98	0.99	0.97	0.98

**Table 5 materials-14-05511-t005:** Calculated parameters for the pseudo-first and pseudo-second order equations.

	Equation Type	R^2^	A, mg g^−1^	k_1_, min^−1^	k_2__,_ mg g^−1^ min^−1^
Se-init	PFO	0.999	54.7 ± 5.5	0.003 ± 0.0002	-
Se-35		0.932	62.2 ± 3.7	0.015 ± 0.003	-
Se-50		0.953	60.1 ± 4.7	0.010 ± 0.002	-
Se-65		0.931	62.7 ± 3.8	0.012 ± 0.002	-
Se-init	PSO	0.853	28.9 ± 7.7	-	0.80 ± 0.04
Se-35		0.998	62.2 ± 1.5	-	9.50 ± 0.80
Se-50		0.999	62.4 ± 2.4	-	7.80 ± 0.60
Se-65		0.998	64.8 ± 2.5	-	6.60 ± 0.50

**Table 6 materials-14-05511-t006:** Degree of mechanical destruction of the materials.

Material	Se-init	Se-35	Se-50	Se-65
Degree of destruction%	8.1	5.2	0.8	0.6

## Data Availability

Data available on request due to restrictions eg privacy or ethical.

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
