# Peer review of "Recovery of Uranium by Se-Derivatives of Amidoximes and Composites Based on Them"

_materials, 2021, doi:10.3390/ma14195511_

Round 1

Reviewer 1 Report

Dear Authors,

I appreciate your laboratory work, but the manuscript ID materials-1331632 entitled „ Recovery of Uranium by Se-Derivatives of Amidoximes and Composites Based on Them” needs strong correction before possible publication. Below you can find my comments.

1) The Introduction describes the features of several materials using in U removal (in general). However, what about materials for UO22+ removal?

2) Lines 119-122, The sentence “It was shown that the introduction of heteroelements with high coordination ability promoted the formation of polymer materials with developed structure and high content of ion-exchange groups (amino-and hydroxyl groups)” sounds like a conclusion and not the aim or scope of the work.

3) line 59: In the sentence „low kinetics of ion measurement” is “measurement” the appropriate word? I don’t understand the part. Maybe “low kinetics of ion adsorption”?

4) I would like to highlight the following article doi:10.3390/molecules24132470. Maybe it will be helpful.

5) Reference no. 28 is invalid. In the article don’t exist following topics: polyethylene (line 100), polyurethane foam (line 103). Please revise the cited sources.

6) line 173: 1000m ???/g, maybe mL

7) lines 176-177: Please describe, what was the reason for the 24-h soaking of sorbents? Hydration, degassing or other?

8) You use two terms interchangeably U(VI) and UO22+. I’m not sure it is ok when you think about the same species (component). If different types of uranium compounds are described by these symbols, please inform in the text.

9) Have the methods described in chapters 2.1-2.3 been adapted from other works or are invented by the Authors? Please provide references and the original elements of the recipes.

10) lines 179-180: Please give the analytical wavelength, the range of detection, etc. I propose because the source [33] isn’t widely available.

11) You sometimes wrote “ml” and sometimes “mL”; two methods of description was used: a/b and a∙b-1. Please unify it.

12) lines 189-192. In my opinion, it would be better to compare at the same concentration in eq/L not in moles/L, because the cations and the anions have different valences. What do you think about it? Of course, I don't expect you to repeat the tests. However, it can change the explanation in lines 361-368.

13) line 205: “V/m = 200 mg/L” it’s mistake

14) Why the equilibrium tests were carried out at pH 6 and 8, and the kinetics at pH 6? The same SEC dependence of cations (pH 6) and anions (pH 6 and 8).

15) The fitting was validated only by R2. It is insufficient and I propose to apply an additional error function. Also, in the methodology, the explanation of the validation way and the calculating procedure is missing.

16) You present two IR spectra. Can you compare these spectra in the description? More clearly indicate the changes between the two materials.

17) Figure 3: Are the resolutions properly connected with the pictures?

18) Could you create subsections in chapter 3? It will be helpful for the readers.

19) Figure 7. If you present curves 1 and 2 it seems logical to add a curve 1-2 to illustrate the actual value of adsorption.

20) text in lines 339-349: 

- The explanation of protonation and deprotonation is evident, however, the Authors didn’t present pHpzc value of synthetized material. This makes it impossible to reliably analyse the data in Figure 7. If the pHpzc test has not been performed, maybe the literature data will make it possible to approximate this value.

- The explanation should be broadened by analysis of uranium(VI) speciation in solution at each pH. This is especially important because the text in present form is too simple.

21) What is the reason for the different types of charts in Figures 8 and 9? The figures represent the same type of data.

22) line 383: anions, not cations

23) line 386: rather [UO2(CO3)3]4-  than [UO2(CO2)3]4- , at pH 6-8 possible is also (UO2)2CO3(OH)3-. However, it depends on pH and concentration and on anion type. What about complexes with phosphates, nitrate and chloride?

24) You forgot to write the value of temperature in the equilibrium test (ISOTHERMS), and also the kinetics and SEC tests.

25) Figure 11: the axle signatures are imprecise: OX - Equilibrium concentration of UO22+, mg/L; OY- Adsorbed UO22+, mg/g (not "adsorbet")

26) lines 412-414: At pH 6 and 8 probably different forms of U complexes are adsorbed.

27) I’m not sure the Se-init series at Fig. 11 b is consistent with the data in table 4.

28) line 426: Of course the silica gel in equivalent value should be tested in your research (equilibrium and kinetic).

29) Why the equilibrium results are in mg/g and kinetic results in mmol/g? It should not be like that. Overall, it is better to model the kinetics and equilibrium for the mg/g data.

30) Figures 13c and 13d are the same photography.

31) Wrong unit of k2 and no unit of k1

32) Probably, the analysis of the results points is incorrect. The Authors fitted all results points from the kinetic experiment. However, some points from the equilibrium area should be excluded when experimental results are fitted by kinetic models. It is well known: Tran, H.N.; You, S.-J.; Hosseini-Bandegharaei, A.; Chao, H.-P. Mistakes and inconsistencies regarding adsorption of contaminants from aqueous solutions: A critical review. Water Res. 2017, 120, 88–116; Moussout, H.; Ahlafi, H.; Aazza, M.; Maghat, H. Critical of linear and nonlinear equations of pseudo-first order and pseudo-second order kinetic models. Karbala Int. J. Mod. Sci. 2018, 4, 244–254

33) The application of IPD model can bring interesting results for comparing Se-init with the silica composites.

34) Modelling of kinetics and equilibrium is poorly discussed

35) line 448: figure 12c doesn’t exist.

36) Equation 7 and data from table 7 are unintuitive. If the material destroys under stirring better will be present the loss: ((m0-m1)/m1)x100 %

37) lines 479-480: Presented data surprised me. Where did these come from?

38) The discussion in the manuscript is minimal. It should be written.

39) I can see an inconsistency in the field of research. The Se-init is investigated towards different pH and cations and anions presence (results from Fig. 7-9), but the composites with silica gel aren’t. Composites are evaluated by kinetics and equilibrium tests.

Author Response

We thank the Reviewer for his interest in our work, as well as very useful comments.

Question 1:  The Introduction describes the features of several materials using in U removal (in general). However, what about materials for UO22+ removal?

Answer: in the cited works, it is assumed that uranium is already present in the form of uranyl ion, the ionic form of which strongly depends on solution pH, therefore we do not provide additional references.

Question 2:  Lines 119-122, The sentence “It was shown that the introduction of heteroelements with high coordination ability promoted the formation of polymer materials with developed structure and high content of ion-exchange groups (amino-and hydroxyl groups)” sounds like a conclusion and not the aim or scope of the work.

Answer: Thanks for your comment, changes were made.

Question 3:  line 59: In the sentence „low kinetics of ion measurement” is “measurement” the appropriate word? I don’t understand the part. Maybe “low kinetics of ion adsorption”?

Answer: Thanks for the comment, corrections were made.

Question 4:  I would like to highlight the following article doi:10.3390/molecules24132470. Maybe it will be helpful.

Answer: We are grateful to the Reviewer for help. A reference to this work will be given in a our subsequent manuscript.

Question 5:  Reference no. 28 is invalid. In the article don’t exist following topics: polyethylene (line 100), polyurethane foam (line 103). Please revise the cited sources.

Answer: We are grateful for your interest in our work. Corrections have been made.

Question 6:  line 173: 1000m ???/g, maybe mL

Answer: We are grateful for this comment. Corrections have been made.

Question 7:  lines 176-177: Please describe, what was the reason for the 24-h soaking of sorbents? Hydration, degassing or other?

Answer: The materials was preliminarily soaking in a solution without a radionuclide, which is a standard procedure in such experiments to remove air bubbles (degassing).

Question 8:  You use two terms interchangeably U(VI) and UO22+. I’m not sure it is ok when you think about the same species (component). If different types of uranium compounds are described by these symbols, please inform in the text.

Answer: We fully agree with this comment. We made some changes and gave an explanation in the section “2.4 Study of Sorption Characteristics under Static Conditions”. In particular, we conventionally designated UO22+ simply as "uranium".

Question 9:  Have the methods described in chapters 2.1-2.3 been adapted from other works or are invented by the Authors? Please provide references and the original elements of the recipes.

Answer: Synthesis the original 4-aminofurazane-3-carboxamidoxime was carried out according to the synthesis given in [34]. The preparation schemes of Se-init sorbent and composites material based on it, given in Sections 2.2 and 2.3, and were fully developed by our research team. We additionally gave a reference to the Se-init material synthesis patent [35].

Question 10:  lines 179-180: Please give the analytical wavelength, the range of detection, etc. I propose because the source [33] isn’t widely available.

Answer: We agree with the comment, the wavelength values have been added to the manuscript.

Question 11:  You sometimes wrote “ml” and sometimes “mL”; two methods of description was used: a/b and a∙b-1. Please unify it.

Answer: Thanks for the comment. Measurement units were unify.

Question 12:  lines 189-192. In my opinion, it would be better to compare at the same concentration in eq/L not in moles/L, because the cations and the anions have different valences. What do you think about it? Of course, I don't expect you to repeat the tests. However, it can change the explanation in lines 361-368.

Answer: We agree with this comment, which we will take into account in future experiments. We have made additional text changes to the manuscript.

Question 13:  line 205: “V/m = 200 mg/L” it’s mistake

Answer: The error has been corrected.

Question 14:  Why the equilibrium tests were carried out at pH 6 and 8, and the kinetics at pH 6? The same SEC dependence of cations (pH 6) and anions (pH 6 and 8).

Answer: The uranium extraction efficiency in the presence of other (competing) cations at pH 6 and 8 is generally comparable to each other. For this reason, this experiment was carried out only at pH6. The most negative effect on the  uranium extraction efficiency is exerted by anions, the effect of which strongly depends on the solution pH. In kinetic experiments, the efficiency of uranium extraction at pH 8 and pH 6 was comparable, and therefore the results were not presented in the work.

Question 15:  The fitting was validated only by R2. It is insufficient and I propose to apply an additional error function. Also, in the methodology, the explanation of the validation way and the calculating procedure is missing.

Answer: To evaluate the results convergence, each experiment was carried out three times, the discrepancy in the data did not exceed 5%. Confidence intervals (for a confidence level of 0.95) were added on the graphs of uranium sorption. The kinetic parameters of sorption, shown in Table 6, were obtained as a result of nonlinear regression of the experimental data by a pseudo-first and -second order equation, which is also indicated in the "Study of Sorption Characteristics under Static Conditions" section.

Question 16:  You present two IR spectra. Can you compare these spectra in the description? More clearly indicate the changes between the two materials.

Answer: Thanks to the reviewer for the comment, the additions were made to the text.

Question 17:  Figure 3: Are the resolutions properly connected with the pictures?

Answer: The figure caption has been corrected.

Question 18:  Could you create subsections in chapter 3? It will be helpful for the readers.

Answer: We separate the section into two subsection: «Physiochemical properties» and « Sorption selective properties».

Question 19:  Figure 7. If you present curves 1 and 2 it seems logical to add a curve 1-2 to illustrate the actual value of adsorption.

Answer: We agree with the comment. We have modified Figure 7 according the Reviewer comment.

Question 20:  text in lines 339-349: 

- The explanation of protonation and deprotonation is evident, however, the Authors didn’t present pHpzc value of synthetized material. This makes it impossible to reliably analyse the data in Figure 7. If the pHpzc test has not been performed, maybe the literature data will make it possible to approximate this value.

- The explanation should be broadened by analysis of uranium(VI) speciation in solution at each pH. This is especially important because the text in present form is too simple.

Answer: We agree with your comment. We added additional text concerning determining results the point of zero charge (pHpzc). Corrections were added to the "Results and Discussion" section.

Question 21:  What is the reason for the different types of charts in Figures 8 and 9? The figures represent the same type of data.

Answer: This type of chart is easy for readers to understand.

Question 22:  line 383: anions, not cations

Answer: We are grateful to the reviewer for this comment. Additional corrections were added to the manuscript text.

Question 23: line 386: rather [UO2(CO3)3]4-  than [UO2(CO2)3]4- , at pH 6-8 possible is also (UO2)2CO3(OH)3-. However, it depends on pH and concentration and on anion type. What about complexes with phosphates, nitrate and chloride?

Answer: We thank the Reviewer for the comment. We have made corrections to the manuscript.

Question 24:  You forgot to write the value of temperature in the equilibrium test (ISOTHERMS), and also the kinetics and SEC tests.

Answer: The additional changes were made to the manuscript text.

Question 25:  Figure 11: the axle signatures are imprecise: OX - Equilibrium concentration of UO22+, mg/L; OY- Adsorbed UO22+, mg/g (not "adsorbet")

Answer: The additional changes were made to the manuscript text.

Question 26:  lines 412-414: At pH 6 and 8 probably different forms of U complexes are adsorbed.

Answer: We have added additions to the manuscript text on the alleged forms of uranium in the solution.

Question 27:  I’m not sure the Se-init series at Fig. 11 b is consistent with the data in table 4.

Answer: We thank the Reviewer for the comment. We have made corrections to the manuscript.

Question 28:  line 426: Of course the silica gel in equivalent value should be tested in your research (equilibrium and kinetic).

Answer: We have added to the manuscript text the additional results of uranium extraction using unmodified silica gel. As a result, we found that when using unmodified silica gel, the uranium extraction efficiency does not exceed 50%. However, the use of silica gel to synthesis new composite materials leads to an increasing of sorption properties. The research data were given in the manuscript.

Question 29:  Why the equilibrium results are in mg/g and kinetic results in mmol/g? It should not be like that. Overall, it is better to model the kinetics and equilibrium for the mg/g data.

Answer: We thank the Reviewer for the comment. We have made corrections to the manuscript.

Question 30:  Figures 13c and 13d are the same photography.

Answer: We thank the Reviewer for the comment. The figure were corrected.

Question 31:  Wrong unit of k2 and no unit of k1

Answer: The results in shown in the table have been updated.

Question 32:  Probably, the analysis of the results points is incorrect. The Authors fitted all results points from the kinetic experiment. However, some points from the equilibrium area should be excluded when experimental results are fitted by kinetic models. It is well known: Tran, H.N.; You, S.-J.; Hosseini-Bandegharaei, A.; Chao, H.-P. Mistakes and inconsistencies regarding adsorption of contaminants from aqueous solutions: A critical review. Water Res. 2017, 120, 88–116; Moussout, H.; Ahlafi, H.; Aazza, M.; Maghat, H. Critical of linear and nonlinear equations of pseudo-first order and pseudo-second order kinetic models. Karbala Int. J. Mod. Sci. 2018, 4, 244–254

Answer: We are grateful to the Reviewer for the references to the work provided. However, there is no need to remove points in the adsorption equilibrium region, this allows to show the absence of uranium desorption and high stability of the material. We understand that the kinetic equations used assume infinite approximation (rather than reaching) the curve to the adsorption equilibrium point.  In the area of ​​the assumed plateau, when the assumed adsorption equilibrium is reached, the exopential growth of the curve obtained using these models will be extremely small, taking into account that the adsorption equilibrium itself is strongly dependent on a number of factors.   Such factors include the solution composition and its pH, temperature, granulation of the sorbent, etc., which, in theory, can introduce larger errors than a large points number  in the area of ​​the assumed adsorption equilibrium. In general, we agree that pseudo-first and pseudo-second order models need to be used with great care. However, the application of these models allows  to give an approximate estimate of the efficiency and uranium extraction rate   based on specific figures, and not a simple description of the curves in text. It should also be noted that the results obtained are not proof of the mechanism of the uranium binding to the sorbent active sites, but only allow one to give a comparative assessment of the extraction rate.

Question 33:  The application of IPD model can bring interesting results for comparing Se-init with the silica composites.

Answer: We are grateful for the Reviewer's comment. We will use this model in our next work.

Question 34:  Modelling of kinetics and equilibrium is poorly discussed

Answer: We have expanded our discussion of kinetic curves.

Question 35:  line 448: figure 12c doesn’t exist.

Answer: the figures numbering were corrected in the manuscript text.

Question 36:  Equation 7 and data from table 7 are unintuitive. If the material destroys under stirring better will be present the loss: ((m0-m1)/m1)x100 %

Answer: Thanks to the reviewer for the comment. The mistake was corrected.

Question 37:  lines 479-480: Presented data surprised me. Where did these come from?

Answer: Thanks to the reviewer for the comment. The mistake was corrected.

Question 38:  The discussion in the manuscript is minimal. It should be written.

Answer: we are grateful for Reviewer interest in our work. We have added additional results and discussions. We would like to note that this work is not the end of our research on Se-derivatives of amidoximes.

Question 39:  I can see an inconsistency in the field of research. The Se-init is investigated towards different pH and cations and anions presence (results from Fig. 7-9), but the composites with silica gel aren’t. Composites are evaluated by kinetics and equilibrium tests.

Answer: Selenium-amidoxime is a sorption-active component; when it is applied to a matrix - silica gel, the mechanism of uranium binding to active sorption sites does not change. In this case, only the accessibility of active sites changes, since the sorption-active component is deposited on the surface of the matrix with a high specific surface area. The purpose of obtaining a composite is to obtain mechanically strong materials with a high ion exchange rate. For this reason, we did not conduct additional studies at different pH and in the presence of competing cations / anions.

Reviewer 2 Report

The authors present a potentially good work, performed experimental work systematically. The appropriate techniques are always welcome. This paper is worthy of publication in the materials after a minor revision.

1.Keywords should start with capital letters.

  1. p1,line 13-14,when the pH of 6-9 the uranyl should be as the hydrolysis form of uranyl ion, you can change “UO22+” to be " U(VI)." .
  2. p9, line 327 change " Se+4 " to be " Se(IV)."
  3. At different pH the UO22+ was as hydrolysis form of uranyl ion appearance, such as UO2(OH)+, and (UO2)3(OH)5+, while the hydrolysis form of uranyl ion will affect the interaction between adsorbent and metal ions. Could you added this section in the pH effection parts?
  4. There are too many figures and table, some of them can be submitted as supply materials, such as table 4, table 6, table 7……
  5. Reuse and desorption are the very important property of adsorbed materials, does the Se-Derivatives of Amidoximes and Composites have this property? can you add these parts in you manuscript?
  6. Figure 10. Scheme of possible structure of binding of UO22+ to Se-init. Can you do some experiment to make sure the structure of binding of UO22+ to Se-init?

The manuscript could be reconsidered for publication after the min. revisions.

Author Response

Question 1:  Keywords should start with capital letters.

Answer: The keywords have been corrected.

Question 2:  p1,line 13-14,when the pH of 6-9 the uranyl should be as the hydrolysis form of uranyl ion, you can change “UO22+” to be " U(VI)." .

Answer: Thanks for the comment, changes were made according to the recommendation.

Question 3:  p9, line 327 change " Se+4 " to be " Se(IV)."

Answer: Thanks for the comment, changes were made according to the recommendation.

Question 4:  At different pH the UO22+ was as hydrolysis form of uranyl ion appearance, such as UO2(OH)+, and (UO2)3(OH)5+, while the hydrolysis form of uranyl ion will affect the interaction between adsorbent and metal ions. Could you added this section in the pH effection parts?

Answer: Thanks for the comment, this description has been added to the manuscript text.

Question 5:  There are too many figures and table, some of them can be submitted as supply materials, such as table 4, table 6, table 7……

Answer: Unfortunately, we disagree with the comment. These tables do not take up much space in our manuscript and are necessary for the results visual presentation and their discussion.

Question 6:  Reuse and desorption are the very important property of adsorbed materials, does the Se-Derivatives of Amidoximes and Composites have this property? can you add these parts in you manuscript?

Answer: We absolutely agree with this comment of the Reviewer. However, these experiments were not included in the aim of this work. The main task was to develop a method for synthesis of new sorption materials, and study their physicochemical properties, as well as to evaluate their sorption-selective characteristics with respect to uranium(VI).  We understand the importance of study the sorbent reuse ability in sorption-desorption cycles and are grateful for your interest. However, experiments to evaluate the new materials use in sorption-desorption cycles is the aim of subsequent work.

Question 7:  Figure 10. Scheme of possible structure of binding of UO22+ to Se-init. Can you do some experiment to make sure the structure of binding of UO22+ to Se-init?

Answer: Figure 10 shows the proposed mechanisms for the uranium binding with active sites of the non-composite material (Se-init). This mechanisms are based on the literature data on amidoxime sorbents. Figure 5f also shows the results of XPS, with the type of uranium hybridization 4f, which allow making such assumptions.

Reviewer 3 Report

In this work, the authors synthesized Se-derivative of amidoxime through the reaction of oxidative polycondensation of N'-hydroxy-1,2,5-oxadiazole-3-carboximidamide with SeO2. Using silica gel with a content of 35, 50, and 65 wt. % as a matrix, the maximum values of adsorption of UO22+ calculated were 620-760 mg/g and 370 mg/g. This work is methodical and meaningful and the idea is interesting, but there are many questions that need to be examined in detail. So I do not think this manuscript achieves the high standard of Materials.

  • For uranium extraction, the main point of concern is the adsorption capacity at low concentration such as 10 ppm and 3 ppb. But the authors did not mention them.
  • The authors stated “The maximum values of adsorption of UO22+ calculated using the Langmuir equation were 620-760 mg/g and 370 mg/g for the composite and non-composite adsorbents, respectively.”. I was curious about how the non-composite adsorbents adsorbed uranium ions? Based on that premise, the authors should carefully analyze the mechanism of adsorption sites.
  • In the introduction, the author uses 10 paragraphs to explain the previous work. Most of the work is unnecessary and messy. The author should simplify the content. In addition, the authors lack the part to introduce their own work.
  • Some descriptions are inappropriate and there are a lot of grammatical errors in the article.
  • The author should calculate the binding capacity and separation sequence of the coordination water through theoretical simulations.

Author Response

Question 1:  For uranium extraction, the main point of concern is the adsorption capacity at low concentration such as 10 ppm and 3 ppb. But the authors did not mention them.

Answer:  Thanks for the Reviewer for this comment. However, we want to note that at a low/ultralow the radionuclide concentration, the sorbents used should not have a high adsorption capacity, but selectivity with respect to the extracted radionuclide. For most of the known sorbents, the selectivity decreases significantly in the many anions presence (carbonate, bicarbonate, sulfate, chloride  etc.) and at pH above 4. In this work, the main aim was not to study the sorption of uranium depending on its concentration. The aim was to synthesize new material and evaluate its effectiveness. We would also like to note that the adsorption isotherms (Fig. 12) have a characteristic form when the curve initial part has a sharply rising character. Such isotherms can be assigned to the L-type (Giles), which indicates a high affinity of the adsorbent and adsorbent. Based on the results obtained, it can be indirectly assumed that at the low uranium concentrations the sorbents will also be effective.

Question 2:  The authors stated “The maximum values of adsorption of UO22+ calculated using the Langmuir equation were 620-760 mg/g and 370 mg/g for the composite and non-composite adsorbents, respectively.”. I was curious about how the non-composite adsorbents adsorbed uranium ions? Based on that premise, the authors should carefully analyze the mechanism of adsorption sites.

Answer: Thanks to the Reviewer for this question. Silica gel is a matrix/carrier of a composite sorbent for fixing a sorption-active component (amidoxime). Such sorbents are mechanically strong and can be easily separated from the solution. Fig. 10 shows the supposed mechanisms for the uranium binding with active sites using the non-composite material Se-init. These  supposed mechanisms are based on the literature data on amidoxime sorbents, as well as the results of XPS (Fig.5f). A detailed study and the mechanisms modeling of uranium binding on the sorption-active sites of the new sorption materials is the aim of subsequent works.

Question 3:  In the introduction, the author uses 10 paragraphs to explain the previous work. Most of the work is unnecessary and messy. The author should simplify the content. In addition, the authors lack the part to introduce their own work.

Answer: We are thanks to Reviewer for interest in our work. We have made additional text changes based on Reviewer comment. We also want to note that the introduction section includes both a description of an effective method for extracting radionuclides from different liquid media, as well as a small description of existing materials. These sections are important for readers to understand the need to develop new sorption materials. The section that includes a description of materials similar in composition is a little less than half of the whole introduction.

Question 4:  Some descriptions are inappropriate and there are a lot of grammatical errors in the article.

Answer: The text was additionally checked by the advanced level translator.

Question 5:  The author should calculate the binding capacity and separation sequence of the coordination water through theoretical simulations.

Answer: We fully agree that the study of the the uranium  binding mechanisms to the new materials active sorption sites is an important task. We would like to note that the results of such studies cannot be published within of the one article. The description and discussion of the results, approaches to research, including models and software, as well as measurement protocols, etc. may be larger than the current manuscript volume!

Round 2

Reviewer 1 Report

Dear Authors,

Thank you very much for improving the manuscript. Please, try to do it once again.

1) Still both forms, U(VI) and UO22+, are present in the text to express uranium, e.g. lines 181, 183.

2) Your explanation could be in the manuscript: “The uranium extraction efficiency in the presence of other (competing) cations at pH 6 and 8 is generally comparable to each other. For this reason, this experiment was carried out only at pH6. The most negative effect on the  uranium extraction efficiency is exerted by anions, the effect of which strongly depends on the solution pH. In kinetic experiments, the efficiency of uranium extraction at pH 8 and pH 6 was comparable, and therefore the results were not presented in the work.”

3) To the answer to question 34 – I don’t see in the text the extension of discussion about kinetic curves.

best regards

Author Response

Question 1:  Still both forms, U(VI) and UO22+, are present in the text to express uranium, e.g. lines 181, 183.

Answer:  We apologize to the Reviewer for this lack; the changes were made to the manuscript.

Question 2:  Your explanation could be in the manuscript: “The uranium extraction efficiency in the presence of other (competing) cations at pH 6 and 8 is generally comparable to each other. For this reason, this experiment was carried out only at pH6. The most negative effect on the  uranium extraction efficiency is exerted by anions, the effect of which strongly depends on the solution pH. In kinetic experiments, the efficiency of uranium extraction at pH 8 and pH 6 was comparable, and therefore the results were not presented in the work.”

Answer:  We are grateful the Reviewer for valuable advice, the edits were made to the manuscript.

Question 3:   To the answer to question 34 – I don’t see in the text the extension of discussion about kinetic curves.

Answer:  We apologize to the Reviewer for this lack; the changes were made to the manuscript.

Reviewer 3 Report

I do not recommend the publication of this paper, although the author has made some corrections. Testing the adsorption of uranium ion at low concentrations is an important criterion for the performance of adsorbent materials. The author emphasizes that this type of material has good selectivity, and I admit that selectivity is very important, but I think the authors should compare and analyze the adsorption capacity and adsorption performance of uranium ions at low concentrations. 

At the same time, the author does not explicitly point out why the substance silica gel (non-composite adsorbent) has good adsorption performance for heavy metal ions?

The authors claim that there is much to be described in future work. But at this stage, there are many problems in this paper that have not been solved, and I think it is inappropriate to do so.

Author Response

Comment 1: I do not recommend the publication of this paper, although the author has made some corrections. Testing the adsorption of uranium ion at low concentrations is an important criterion for the performance of adsorbent materials. The author emphasizes that this type of material has good selectivity, and I admit that selectivity is very important, but I think the authors should compare and analyze the adsorption capacity and adsorption performance of uranium ions at low concentrations. 

Answer: We agree that the efficiency evaluation of radionuclide extraction can be carried out using the solutions with radionuclide ultra-low concentration. Such studies, for example, is commonly carried out using cesium or strontium radionuclides. The activity of such radionuclides can be easily determined using gamma spectrometry (NaI (Tl) or HPGe based detectors) or liquid scintillation radiometry, respectively. However, in our case, measurement of the uranium trace amounts content in solutions by gamma- or alfa-spectrometry is limited by a low quantum yield, high measurement error, and with requires a number of complex and routine procedures. For this reason, solutions containing macro concentrations of uranium were used in the work; however, this approach also makes it possible to assess the sorbents efficiency. We disagree with the Reviewer that in the ultralow concentrations region it is necessary to evaluate the adsorption capacity for the following objective reasons. In the ultra-low concentrations region , there is an excess of active exchange sites, the number of which significantly exceeds the  uranium amount in solution. The interaction between already adsorbed uranium on the active sites will be minimal, and the "power parameter" (responsible for active sites heterogeneity) in the Freundlich equation will tend to unity, in fact, this will be Henry's equation. It is quite obvious that in this case there is no point to evaluate the adsorption capacity in the ultralow concentrations region. As mentioned in the answers to previous comments, the affinity of the adsorbent and adsorbate can be simply estimated with the Langmuir equation using the "adsorption equilibrium constant". In fact, the almost vertical growth of the initial part of adsorption isotherm indicates an extremely high affinity of the adsorbent and adsorbate. It is also worth noting that the initial part of the Langmuir isotherm in the low concentration region can be described by the Henry equation. This allows the results to be extrapolated to solutions with low concentration.

The recovery efficiency of a radionuclide micro concentration is usually estimated based on the recovery efficiency in percent or the distribution coefficient. At the same time, the distribution coefficient calculated using a known simple equation, in contrast to the recovery efficiency in percent, will be less dependent on the ratio of the solid/liquid phases, therefore, it is more preferable.

We assume, although we are not sure, that the Reviewer is referring to a special case, where uranium trace amounts are presented in the form of a complex multinuclear complex ion, additionally associated with complexing agents, which decrease the uranium efficiency extraction. In this case, we would like to note that the aim of this work was to synthesize new materials and evaluation the sorption characteristics using uranium solution.

In addition, now we are don’t mean the practical application of sorbents, we are not talking about permissible  uranium levels  and the possibility of treatment real solutions using our sorbent, we provide information on the use of this approach and its possibilities for obtaining new class adsorbents.

Comment 2: At the same time, the author does not explicitly point out why the substance silica gel (non-composite adsorbent) has good adsorption performance for heavy metal ions?

Answer: This question was not presented in the first review, so we are answering it now. The mechanisms of adsorption of heavy metals on silica gel have been well studied in the following works: «Da’na, E. Adsorption of Heavy Metals on Functionalized-Mesoporous Silica: A Review. Microporous and Mesoporous Materials 2017, 247, 145–157, doi:10.1016/j.micromeso.2017.03.050», and «Sandoval, O.G.M.; Trujillo, G.C.D.; Orozco, A.E.L. Amorphous Silica Waste from a Geothermal Central as an Adsorption Agent of Heavy Metal Ions for the Regeneration of Industrial Pre-Treated Wastewater. Water Resources and Industry 2018, 20, 15–22, doi:10.1016/j.wri.2018.07.002». To evaluate the effect of silica gel on the composites sorption characteristics, a control experiment was carried out, results of which showed that silica gel in non-modificated form, poorly extracts uranium. It was shown that the efficiency of uranium extraction does not exceed 40%. We also want to note that pure amidoximes are can adsorb of uranium without being applied to silica gel. Silica gel is a matrix with a high specific surface area that increases the mechanical strength of the adsorption material..

Comment 3: The authors claim that there is much to be described in future work. But at this stage, there are many problems in this paper that have not been solved, and I think it is inappropriate to do so.

Answer: Unfortunately, we cannot answer this comment, due to the fact that the Reviewer does not provide specific examples. Therefore, we do not quite understand what problems the Reviewer is talking about. If the Reviewer does not agree that there are no experiments using solutions containing the uranium ultralow concentration, then an extended answer is given above.